# BiTGNN: PREDICTION OF DRUG-TARGET INTERACTIONS BASED ON BIDIRECTIONAL TRANSFORMER AND GRAPH NEURAL NETWORK ON HETEROGENEOUS GRAPH

## ABSTRACT

Drug-target interaction (DTI) is a widely explored topic in the field of bioinformatics and plays a pivotal role in drug discovery. However, the traditional bio-experimental process of drug-target interaction identification requires a large investment of time and labor. To address this challenge, graph neural network (GNN) approaches in deep learning are becoming a prominent trend in the field of DTI research, which is characterized by multimodal processing of data, feature learning and interpretability in DTI. Nevertheless, some methods are still limited by homogeneous graphs and single features. To address the problems we mechanistically analyze graph convolutional neural networks (GCN) and graph attentional neural networks (GAT) in order to propose a new model for drug-target interaction prediction based on graph neural networks named BiTGNN (bidirectional transformer and graph neural network). The method first establishes drug-target pairs through the pseudo-position specificity scoring matrix (PsePSSM) and drug fingerprint data, and constructs a heterogeneous network by utilizing the relationship between the drug and the target. Then, the computational extraction of drug and target attributes is performed using GCN and GAT for the purpose of model information flow extension and graph information enhancement. We collect interaction data using the proposed Bi-directional transformer (Bi-transformer) architecture, in which we design a bi-directional cross-attention mechanism for calculating the effects of drug-target interactions for realistic biological interaction simulations. Finally, a feed-forward neural network is used to obtain the feature matrices of the drug and the target, and DTI prediction is performed by fusing the two feature matrices. The Enzyme, Ion Channel (IC) , G Protein-coupled Receptor (GPCR) , and Nuclear Receptor (NR) datasets are used in the experiments, and compared with several existing mainstream models, our model outperforms the others in Area Under the Curve (AUC), Area Under the Precision-Recall Curve (AUPR) , Accuracy and Specificity metrics.

## 1 INTRODUCTION

Despite significant progress in pharmaceutical research and development, the traditional drug discovery process continues to be plagued by risks, time constraints, and exorbitant costs Paul et al. (2011); Adams & Brantner (2006). Currently, a key approach to expediting drug discovery involves the discernment of potential interactions between drugs and their respective targets Nunez et al. (2012). This identification process plays a critical role in efficiently screening novel drug candidates Chen et al. (2018). The exponential growth of drug and target data has led to an increase in drugs without corresponding target information, rendering traditional experiments time-consuming and labor-intensive. As a result, researchers are increasingly adopting machine learning and deep

learning techniques to construct DTI prediction networks. This progressive trend in DTI research has contributed to the identification of new DTIs, thereby promoting the development of combination drugs. In addition, DTI prediction is crucial for any repositioning of existing drugs Langedijk et al. (2015).

In recent years, traditional machine learning (ML) algorithms have been used to model the prediction of interactions between compounds and proteins as a binary classification problem Batouche & Bahi (2021); Yuan et al. (2016); Olayan et al. (2018); Mohamed et al. (2019). With the rapid development of deep learning, many variants of the GNN model have been proposed to achieve state-of-the-art performance Nguyen et al. (2021); Long et al. (2019); Peng et al. (2021); Rifaioglu et al. (2020); Tsubaki et al. (2019); Wan et al. (2019); Zheng et al. (2020). It is then concluded that in the field of drug-target interaction analysis, various methods can be categorized based on the method of extracting features from the drug and target. These approaches fall into two primary groups: one relies on independent features of drug-target pairs, while the other leverages interactive information Cheng et al. (2022). The former category employs an independent feature extractor to obtain distinct feature vectors for both drugs and targets.

Despite the commendable performance achieved by the methods relying on the independent characteristics of drugs and targets, it is crucial to acknowledge that drug-target interaction encompasses the interplay of drug-target pairs within a high-dimensional space Schenone et al. (2013). Consequently, combining interactive information concerning drug-target pairs for predictive purposes becomes highly rational. The approach based on interaction information takes into account authentic biological drug-target interaction processes and effectively simulates them. Notably, drugs and targets are no longer treated as discrete elements within the feature extraction process. As an illustration, Chen et al. (2020) introduced a transformer-based neural network called Transformer-CPI, which effectively mitigates certain particular shortcomings encountered in sequence-based DTI models.

This investigation adopts a methodological approach entailing the extraction of amino acid sequences from diverse protein classifications, encompassing Enzyme, IC, GPCR, and Nuclear Receptor NR. The principal aim is to proficiently apprehend the inherent characteristic information embedded within these amino acid sequences. To realize this objective, PsePSSM features are employed, facilitating the encoding of both evolutionary and sequential insights pertaining to the protein sequences. Regarding drug characterization, this article employs molecular "fingerprints" as essential inputs for drug characterization due to their pivotal role in chemoinformatics and machine learning within drug discovery applications Kearnes et al. (2016).

Building upon the work of Chen et al. (2020) in capturing potential features, we propose BiTGNN. Recognizing the heterogeneity of the DTI graph and the varying significance of different neighbors, we leverage the graph attention layer to explore the attention weights of adjacent nodes, thereby significantly enhancing model capacity. In this paper, we present BiTGNN as a solution to the aforementioned challenges. It fuses the interactive and independent features between drugs and targets to predict DTIs. During this process, GNN and GAT are selected as independent feature extractors, capable of capturing abundant semantic information associated with drugs and targets. Following this, the feature vectors are fed into a Bi-transformer, which considers the local substructures of both the drug and target, facilitating the extraction of interaction features for downstream classification. Finally, the fused feature vectors are inputted into a fully connected layer to predict DTIs. Experimental results demonstrate that BITGNN outperforms other state-of-the-art methods across four distinct datasets, as evidenced by superior AUC, AUPR, precision, and accuracy metrics.

In summary, the contributions of this paper are summarized as follows: to the best of our knowledge, this is the first time that a Bi-transformer has been considered to take into account drug-target interaction features; We utilize GAT and GCN as drug-independent feature extractors to extract their semantic information for targets as well; Our comprehensive experimental results show that the method outperforms the four state-of-the-art methods on the four benchmark datasets.

## 2 MODEL

### 2.1 PROBLEM DESCRIPTION

Target dataset $P = \{p_1, p_2, \cdots, p_m\}$ and drug dataset $D = \{d_1, d_2, \cdots, d_r\}$ are given, containing $m$ targets and $r$ drugs respectively. Our goal is to predict whether there will be an interaction between the target $p_i$ and the drug $d_j$. A binary matrix indicates whether there is an interaction between the drug and the target. If $A_{pd}(i, j) = 1$, it indicates that the target interacts with the drug; otherwise $A_{pd}(i, j) = 0$.

### 2.2 OVERVIEW

In this study, we propose a new method called BiTGNN to predict DTIs. We first briefly describe the method and then focus on the different modules of the method. Figure $1(a)$ shows the network architecture of BiTGNN. It consists of four modules: the graph construction module, the node feature aggregation module, the Bi-transformer module, and the prediction module. The node feature aggregation module and the Bi-transformer module are the core of the BiTGNN model. The Bi-transformer is composed of self-attention, bidirectional cross-attention, feedforward neural network, and multi-layer perceptron. Using the Bi-transformer module, we can extract the interactive information of drugs and targets.

PsePSSM and fingerprint are used as feature representations for given targets and drugs, respectively. We construct a network of drugs and targets, where node represents drug or target, and edge represents interactions between drugs and targets, interactions between drugs and drugs, and interactions between targets and targets. We then apply graph attention networks and graph convolutional networks to generate drug and target embeddings in the heterogeneous network. It is worth noting that different neighbors have different importance. In implementation, the attention mechanism is used to assign different weights to different neighbors, which will increase the capacity of the model. A Bi-transformer is used to extract the interaction features of drugs and targets in subsequent sessions. Through bidirectional cross-attention, drug features are considered while learning target features, and target features are considered when learning drug features. The BiTGNN model has obtained better results.

### 2.3 GRAPH CONSTRUCTION

**Target feature representation** According to the previous research setup by Shi et al. (2018), protein sequences are represented as a PsePSSM features to encode the evolution and sequence information of proteins with different length sequences. The settings for this paper are the same as before Shi et al. (2018).

For a target sequence $p_m$ with $L$ amino acid residues, we use the position-specific scoring matrix (PSSM) as its descriptor introduced by Jones (1999). The PSSM with a dimension of $L \times 20$ can be expressed as:

$$A_{PSSM} = \begin{bmatrix} E_{1 \to 1} & E_{1 \to 2} & \cdots & E_{1 \to j} & \cdots & E_{1 \to 20} \\ E_{2 \to 1} & E_{2 \to 2} & \cdots & E_{2 \to j} & \cdots & E_{2 \to 20} \\ \vdots & \vdots & \vdots & \vdots & \vdots & \vdots \\ E_{i \to 1} & E_{i \to 2} & \cdots & E_{i \to j} & \cdots & E_{i \to 20} \\ \cdots & \cdots & \cdots & \cdots & \cdots & \cdots \\ E_{L \to 1} & E_{L \to 2} & \cdots & E_{L \to j} & \cdots & E_{L \to 20} \end{bmatrix}$$

Here, $j$ is the natural amino acid types, and $E_{i \to j}$ is the score of the $i$-th residue in the amino acid sequence mutated to the $j$-th amino acid residue, which can be searched using PSI-BLAST in the Swiss-Prot database Altschul et al. (1997).

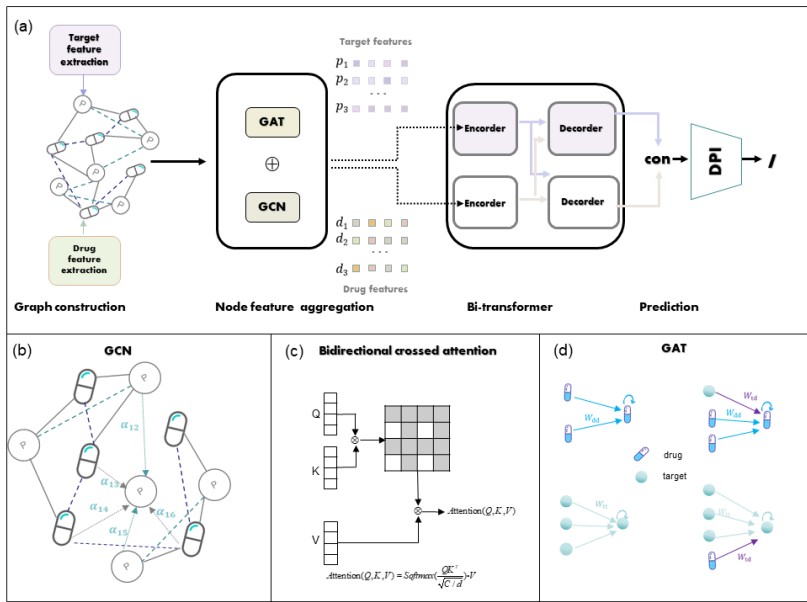

Figure 1: (a) The graph is the general framework of the model. It is divided into four parts from left to right, including the graph construction module, the node feature aggregation module, the dual converter module, and the prediction module. (b) The figure shows the schematic diagram of GCN in a heterogeneous network. (c) Fig. shows the schematic diagram of GAT in a heterogeneous network, from left to right (1), (2), (3), and (4). Figure (d) shows bidirectional cross-attention. The self-attention block receives input from a single source and the cross-attention block receives information from two sources.

The following equation is used to normalize matrix elements to intervals:

$$\bar{E}_{i \to j} = \frac{1}{1 + \exp\left(-E_{i \to j}\right)}. \tag{1}$$

However, according to the PSSM descriptor, the number of amino acids in different proteins varies, so the number of rows in the PSSM matrix varies. One possible method to make PSSM descriptors a unified representation is to represent protein sample $P$ as:

$$\bar{A}_{PSSM} = \left[\bar{E}_1, \bar{E}_2, \ldots, \bar{E}_{20}\right]^T \tag{2}$$

Here $T$ is the transpose operator.

$$\bar{E}_j = \frac{1}{L} \sum_{i=1}^{L} \bar{E}_{i \to j} \tag{3}$$

where $\bar{E}_{i \to j}$ is the score of the residue of the $i$-th position in the amino acid sequence changed to the $j$-th amino acid residue after normalization, $\bar{E}_j$ is the average score of the amino acid residue in protein $p$ being mutated amino acid type $j$ during the process of evolution. However, if Eq. (2) is used to represent protein $p$, all sequence information will be lost. To avoid complete loss of sequence information, in combination with the concept of the pseudo-amino acid composition (PseAAC) initially proposed by Chou (2001), we use PsePSSM to represent the protein $p$

$$P^{\lambda}_{\text{Pse}PSSM} = \left[\overline{E}_1, \overline{E}_2, \cdots, \overline{E}_{20}, G_1^1, G_2^1, \cdots, G_{20}^1, \cdots, G_1^{\lambda}, G_2^{\lambda}, \cdots, G_{20}^{\lambda}\right]^T \tag{4}$$

where,

$$G_j^{\lambda} = \frac{1}{L-\lambda} \sum_{i=1}^{L-\lambda} \left[\bar{E}_{i \to j} - \bar{E}_{i+\lambda \to j}\right]^2 \ (j = 1, 2, \cdots 20; 0 \leq \lambda \leq L) \tag{5}$$

where $G_j^\lambda$ is the correlation factor of the $j$-th amino acid and the continuous distance along the protein sequence is $\lambda$. This means that $G_j^1$ represents the relevant factor coupled along the most continuous PSSM score on the protein chain of amino acid type $j$, $G_j^2$ represents the second closest PSSM score by coupling, and so on. Therefore, a protein sequence can be expressed as Eq. (4) using PsePSSM and generates a $(20+20\lambda)$-dimensional feature vector.

The PsePSSM algorithm converts the protein sequences with different lengths in the dataset into vectors with the same dimension after feature extraction. In this paper, $\lambda$ is set to 10 after executing the optimization program for the training sample by 5-fold cross-validation. So, to facilitate the implementation of the following algorithm, the characteristic dimension of each target is 220.

**Drug feature representation** Some researchers commonly use drug fingerprints to represent drug features, such as Sun et al. (2021) and Nguyen et al. (2023), who have shown that using fingerprints to represent drug features can improve model performance. For drug molecules, we used the chemical structure of molecular substructure fingerprints from the PubChem database Kim (2019). For each drug molecule, it defines an 881-dimensional binary vector $Q$ to represent the molecular substructure, where the corresponding bit code of the vector is 1s, indicating that the substructure exists, and the code of nonexistence is 0s. Therefore, given a drug, its fingerprint features are represented as

$$Q\left(d_n\right) = \left[q_1\left(d_n\right), q_2\left(d_n\right), \cdots, q_{881}\left(d_n\right)\right]. \tag{6}$$

**Construct heterogeneous graphs** Given a dataset of $m$ targets $P = \{p_1, p_2, \ldots, p_m\}$ and $r$ drugs $D = \{d_1, d_2, \ldots, d_r\}$, where $p_i \in R^p, d_i \in R^d$, where $p$ and $d$ represent the dimensions of PsePSSM and fingerprint, respectively. The interactions between drugs and targets are represented by $T = (t_1, t_2, \ldots, t_k)$, $t \in \{1, -1\}$, where 1 means the relationship between two nodes is pulled in and -1 means two nodes are distant. In the drug-target interaction bipartite graph, the bipartite graph is transformed into a heterogeneous graph by increasing the similarity between drugs and the similarity between proteins.

We make additive aggregation easier by unifying the features of PsePSSM and drug fingerprints into same dimension. This paper uses two weighting matrices to achieve this goal Wang et al. (2021)

$$\begin{aligned} p_i' &= ReLU\left(W_p \cdot p_i\right), \quad W_p \in R^{p \times F}, \quad p_i \in R^F, \\ d_i' &= ReLU\left(W_d \cdot d_i\right), \quad W_d \in R^{p \times F}, \quad d_i \in R^F. \end{aligned} \tag{7}$$

We combine the new target dataset $P' = (p_1', p_2', \cdots, p_m')$ and the new drug dataset $D' = (d_1', d_2', \cdots, d_r')$ with the graph nodes $H = (h_1, h_2, \cdots, h_K), h_i \in R^F$. These nodes are connected to the undirected edge $e \in \{1, -1\}$, indicating the similarity between drug-target interactions, drug-drug interactions, and target-target interactions.

The edge set, $E$, contains two different components: interaction edges and similarity edges. The interaction edge, $e_i$, is either 1 or -1, referring positive relation and negative relation of two nodes. Since the positive relation is from the ground truth interaction targets set, $T$, the initial graph of the DTI is very sparse and has an imbalanced issue. To solve the above issues, the negative samples are selected randomly from the unidentified drug-target pairs. We assume that the positive samples are a small percentage of all possible samples, so there is a low probability that real interaction be selected as a negative sample. The proportions of positive samples detected in each dataset are 0.99% (Enzymes), 3.49% (ICs), 2.99% (GPCRs), and 6.40% (NRs). In the experiment, we choose a negative sample with the same number of positive samples. However, the initial interaction edges are the edges between the drugs and the targets, drugs and targets construct a bipartite graph, which limits the information flow. So, by adding edges between drugs and targets, the bipartite graph is converted into a heterogeneous graph. Similar edge $e$ is based on DTI bipartite graphs and their common neighborhood information. If the number of common positive or negative neighbors of two nodes is greater than the threshold $\theta$, the two nodes are represented by 1, which means they are similar, Otherwise, they are represented by 0.

### 2.3.1 NODE FEATURE AGGREGATION MODULE

**GCN on heterogeneous graphs**

In this study, the graph convolution module uses adjacent nodes of the central node in graph $G$ to define the information propagation framework. These parameters and weights are shared across all local computational graphs, and the same information propagation method should be used in the same local computational graph. As shown in the GAT diagram in Figure 1($d$), there are four different local calculation diagrams: (1), (2), (3), and (4). In (1), the central node is drug $d_1$, and all adjacent nodes are drugs; In (2), the central node $d_3$ is the drug, and there are two adjacent nodes: drug $d_1$, $d_5$, and target $t_4$. (3) and (4) are the other two cases where the target node is in the center. Add the features of the same drug node calculated by (1) and (2) to obtain its embedded representation. Similarly, according to (3) and (4), the feature representation of the target node can be obtained. The calculation method for node embedding is as follows

$$h'_d = h_d^{\prime(1)} + h_d^{\prime(2)}, h'_p = h_p^{\prime(3)} + h_p^{\prime(4)} \tag{8}$$

where $h'_d$ represents the embedding representation of drug node $d$; $h_d^{\prime(a)}$ and $h_d^{\prime(b)}$ represent the hidden states of node $d$ in the local calculation graphs ($a$) and ($b$), respectively; $h'_p$ represents the embedding representation of the target node $p$ and $h_p^{\prime(c)}$ and $h_p^{\prime(d)}$ represent the hidden states of the node $p$ in the local calculation graphs ($c$) and ($d$), respectively.

In each layer of GCN, four local computational graphs are calculated based on the types of edges in the original graph to propagate and aggregate node information. The aggregation method for single-layer graph convolution is as follows

$$h_i^{\prime(t+1)} = \delta \left( \sum_{\tau} \sum_{j \in \mathcal{N}_t^i} W_\tau^{(t)} h_j^{\prime(t)} \right) \tag{9}$$

where $h_i^{\prime(t)} \in \mathbb{R}^{d^{(t)}}$ represents the hidden state of node $i$ in the $k$-th layer of the GCN, and $d^{(t)}$ represents the dimension of node embedding in the $t$-th layer. $\tau$ represents the type of edge in the heterogeneous graph $G$, such as drug-drug, target-target and drug-target. $W_\tau^{(k)}$ is the weight of edge type $\tau$ in the $k$-th layer, and the weight of the same edge type is shared. $\mathcal{N}_\tau^i$ represents the set of direct neighbors of node $i$ under type $\tau$, including $i$ itself. $\delta$ is the $ReLU$ activation function. As shown in Figure 1(b).

**GAT on heterogeneous graphs** Our approach uses GAT Velikovi et al. (2017) to adaptively learn weights for each edge and represent each node by message passing. The input of GAT is a set of node features $H = (h_1, h_2, \cdots, h_K), h_i \in R^F$ and DTI adjacent matrix $A$. $H$ contains $H_p = (h_{p_1}, h_{p_2}, \ldots, h_{p_m})$ and $H_d = (h_{d_1}, h_{d_2}, \ldots, h_{d_r})$ and $A$ is generated by the construct heterogeneous graphs. The output is a new set of node features $H'' = (h''_1, h''_2, \cdots, h''_K), h''_i \in R^{F''}$

To convert the input feature to a higher level feature, we apply a weight matrix, $W \in R^{F' \times F}$, to each node.

$$f_i = \sigma \left( W \cdot h_i \right), f_i \in R^{F'}. \tag{10}$$

Then we perform a self-attention mechanism on node pair, $a: R^{F'} \times R^{F'} \to R$, to compute attention weight, which indicates the importance of $n_j$ to $n_i, n_j \in \mathcal{N}_i$, where $\mathcal{N}_i$ is the neighborhood of $n_i$ and itself in DTI graph.

$$w_{ij} = a \left( f_i, f_j \right), w_{ij} \in R^1. \tag{11}$$

In the GAT layer, the attention mechanism $a$ is a single-layer neural network, parametrized by a weight matrix $\tilde{a} \in R^{2F'}$. Then the LeakyReLU nonlinearity is applied.

$$w_{ij} = LeakyReLU \left( \tilde{a}^T \left[ f_i \| f_j \right] \right) \tag{12}$$

where $T$ is transposition and $\|$ is the concatenation operation. In the general formulation, the attention mechanism allows every node to attend to every other node, dropping all structural information. To add graph structure information, we perform mask attention according to the DTI adjacent matrix $A$, which enables only the neighbor nodes to be attended. Then we normalize the attention weight across all choices of $j$ using the softmax function to make it comparable to different nodes.

$$\alpha_{ij} = softmax\,{\rm x_j}(w_{ij}) = \frac{\exp\left(w_{ij}\right)}{\sum_{k \in \mathcal{N}_i} \exp\left(w_{ik}\right)} \tag{13}$$

where $\mathcal{N}_i$ is the set of $i\text{'}s$ neighborhood and itself. After obtaining the normalized attention score, we use message passing to compute a linear combination of the node features and output the final aggregated features of each node.

$$h_i'' = \sigma\left(\sum_{j \in \mathcal{N}_i} \alpha_{ij} W h_j\right) \tag{14}$$

where $\sigma$ is a nonlinearity, $H''$ is final aggregated features.

Finally, based on the outputs of GCN and GAT, the feature representation of node i is as follows

$$q_i = h_i' + h_i'' \tag{15}$$

where $h_i'$ and $h_i''$ are the outputs of GCN and GAT respectively. Here $i$ denotes the drug and target serial number.

### 2.3.2 BIDIRECTIONAL TRANSFORMER

**Eecoder** The encoder is mainly composed of self-attention, layer normalization, feedforward neural networks, and residual blocks. Its inputs are three matrices $V, K, Q$, where $Q \in R^{l_q \times d_k}, K \in R^{1_k \times d_k}, V \in R^{1_v \times d_v}, l_{\rm q}, l_k, l_v$ is the dimension of the input length, $d_k$ and $d_v$ is the dimension after conversion, and then the output matrix is

$$\text{Attention}(Q, K, V) = softmax\left(\frac{QK^T}{\sqrt{d_k}}\right) V \tag{16}$$

where $Q, K$ and $V$ are created from the output of the GAT by the projection functions $f = W^T x + b$ (where $W$ and $b$ are weight and bias, respectively).

**Decoder** The decoder is mainly composed of bidirectional cross-attention, feedforward neural network, and layer normalization. After taking the outputs from the encoder, we need to precisely integrate them to capture valuable drug and target features and reveal the properties of the DTI. These information sources are increasingly valued because many researchers have demonstrated that this multi-modal feature can improve model performance. $K$ and $V$ of the targets are used as inputs in one direction of cross-attention, and the $Q$ matrix of the drug is used as input in the other direction of cross-attention. Our goal in the cross-attention block is to force the model to capture patterns that show the effect of information from the compound on the target information and the effect of information from the target on the compound information.The final interaction representation can be expressed as follows

$$Interactions = CrossAttention(V, K, Q) \tag{17}$$

The $Q, K$ and $V$ from the encoder are generated by $f = W'x + b$, where $W$ and $b$ are weight and bias, respectively. We found that a single-headed cross-attention is superior to other multi-headed cross-attention. As shown in Figure 1($c$).

### 2.3.3 THE FINAL DECODER

The final decoder is a neural network, parametrized by a weight matrix $W \in R^{2F'}$. It takes pairs of drug-protein embeddings, generated by the Bi-transformer layer (e.g. $q_i'$ and $q_j'$), as input. Then the two node vectors do an element-wise multiplication, $p \odot d \to v, v, p, d \in R^F$. Finally, through a layer of neural network $v^F \to R^1$ and a Sigmoid activate the function, produce a probability score indicating whether they interact:

$$s_{ij} = Sigmoid \left( ReLU \left( W \left( q_i' \odot q_j' \right) \right) \right).$$

## 3 EXPERIMENTAL SETTING

### 3.1 DATASETS

The Yamanishi dataset contains four sub-datasets: Enzyme, IC, GPCR, and NR datasets. Each sub-dataset contains three networks: drug-drug structure similarity network, target–target similarity network, and DTI network. As a widely used dataset, the Yamanishi dataset makes it easier for researchers to compare their algorithms with state-of-the-art methods.

Table 1: The evaluation results of BiTGNN and other baseline methods on Enzyme's dataset.

|      | RF | SVM | GCN | GAT | DTIGAT | DTICNN | DTIGNN | EEGDTI | SGCLDTI | MHADTI | BiTGNN |
|------|------|------|------|------|------|------|------|------|------|------|------|
| AUC  | 0.8202 | 0.7886 | 0.7594 | 0.8485 | 0.9627 | 0.9335 | 0.9132 | 0.9001 | 0.9315 | 0.9440 | **0.9730** |
| AUPR | 0.8351 | 0.8116 | 0.7970 | 0.8335 | 0.9613 | 0.9338 | 0.8488 | 0.8436 | 0.9177 | 0.9373 | **0.9796** |

The best result is indicated in bold, and the second best result is marked with an underline.

Table 2: The evaluation results of BiTGNN and other baseline methods on GPCR's dataset.

|      | RF | SVM | GCN | GAT | DTIGAT | DTICNN | DTIGNN | EEGDTI | SGCLDTI | MHADTI | BiTGNN |
|------|------|------|------|------|------|------|------|------|------|------|------|
| AUC  | 0.8423 | 0.8009 | 0.7658 | 0.7753 | 0.7622 | 0.8543 | 0.8634 | 0.8793 | 0.8787 | 0.8814 | **0.8819** |
| AUPR | 0.8502 | 0.8534 | 0.7676 | 0.7680 | 0.7649 | 0.8510 | 0.8552 | 0.8432 | 0.8720 | 0.8596 | **0.9125** |

The best result is indicated in bold, and the second best result is marked with an underline.

Table 3: The evaluation results of BiTGNN and other baseline methods on IC's dataset.

|      | RF | SVM | GCN | GAT | DTIGAT | DTICNN | DTIGNN | EEGDTI | SGCLDTI | MHADTI | BiTGNN |
|------|------|------|------|------|------|------|------|------|------|------|------|
| AUC  | 0.8402 | 0.8200 | 0.7947 | 0.9084 | 0.7867 | 0.8918 | 0.8879 | 0.8960 | 0.8989 | 0.9173 | **0.9695** |
| AUPR | 0.8229 | 0.8199 | 0.8156 | 0.8816 | 0.7778 | 0.8890 | 0.8320 | 0.8553 | 0.8883 | 0.8948 | **0.975** |

The best result is indicated in bold, and the second best result is marked with an underline.

### 3.2 BASELINE

As a comparison, we use RF, SVM, TriModel, GCN, GAT, DTIGAT, DTICNN, DTIMGNN, EEGDTI, MKTCMF, MHADTI, CnnDTI, PSSMLPQ, CNNEMS, PreDTIs, DTI-HETA to compare with our model to demonstrate the feasibility of our model. Since DTI prediction is a categorical task, we use accuracy, precision, the AUC, and AUPR as indicators to measure model performance. The best results are shown in bold, and the second best are underlined.

### 3.3 EXPERIMENTAL RESULT

In this study, a 5-fold cross-validation (CV) method is used to carry out experiments. 5-fold CV can effectively reduce random errors in model evaluation and improve the accuracy of evaluation results. At the same time, it can also make full use of datasets and reduce the deviation introduced by unreasonable data segmentation. Therefore, 5-fold CV is one of the most commonly used model evaluation methods.

Table 4: The evaluation results of BiTGNN and other baseline methods on NR's dataset.

|  | RF | SVM | GCN | GAT | DTIGAT | DTICNN | DTIGNN | EEGDTI | SGCLDTI | MHADTI | BiTGNN |
|---|---|---|---|---|---|---|---|---|---|---|---|
| AUC | 0.8400 | 0.8378 | 0.7049 | 0.8204 | 0.9120 | 0.7333 | 0.8603 | 0.8778 | **0.9323** | 0.9099 | 0.8673 |
| AUPR | 0.8323 | 0.8196 | 0.7553 | 0.8082 | 0.9032 | 0.7447 | 0.8429 | 0.8655 | 0.8994 | 0.9150 | **0.9153** |

The best result is indicated in bold, and the second best result is marked with an underline.

Table 5: The evaluation results of BiTGNN and other baseline methods on Enzyme's dataset.

|  | CnnDTI | Zhao et al. | PSSM+LPQ | CNNEMS | PreDTIs | DTI-HETA | Bi-TDTI |
|---|---|---|---|---|---|---|---|
| Accuracy | 0.943 | 0.9032 | 0.8915 | 0.9419 | 0.9067 | **0.94702** | 0.93162 |
| Specificity | N/A | N/A | N/A | N/A | 0.8578 | 0.9222 | **0.97298** |

The best result is indicated in bold, and the second best result is marked with an underline.

Many prediction methods have been proposed for DTI prediction. Tables 1 through 2 detail how BiTGNN compares to other baseline methods on different datasets. Specifically, Table 1 details the performance of BiTGNN and other methods on Enzyme data. The AUC and AUPR scores of the best-performing BiTGNN are 97.30% and 97.96%, respectively, which are 0.0103 and 0.0183 higher than that of the second-best model because our model uses cross-attention to consider the effect of drugs or targets on targets or drugs. Table 2 details the performance of BiTGNN and other methods on GPCR data. BiTGNN shows the best performance in AUC and AUPR, which are 0.0005 and 0.0405 higher than the second-best model, although MHADTI uses multi-source drugs and targets to construct its similarity network, SGCL uses GAT to extract features. Similarly, Table 3 details the performance of BiTGNN and other methods on the IC dataset. The AUC and AUPR values of BiTGNN are 0.0522 and 0.0802 higher than those of MHADTI, the second-best effect. As for the NRs dataset, our model is lower in AUC than the best-performing DTIGAT and slightly higher in AUPR than MHADTI. This is mainly since the NR dataset only contains a very limited number of samples for training, and all the deep learning-based methods suffer from it.

Due to the small sample size of NRs and GPCRs, we only used Enzyme and IC datasets when comparing acc and spec. The overall performance of our model BiTGNN is excellent. Specifically, the performance of BiTGNN on Enzyme data is shown in Table 5. The acc of BiTGNN is slightly lower than that of CNNEMS, and BiTGNN scores 97.298% in terms of spec, 0.05078 higher than the next best model, DTI-HETA. Table 6 compares BiTGNN with other methods on the IC dataset. Our model BiTGNN performs best, with an acc and spec of 0.00344 and 0.02686 higher than the next most effective model, respectively.

## 3.4 ABLATION EXPERIMENT

We remove the GAT module, the GCN and Bi-transformer modules, Bi-transformer module modules from the enzyme dataset separately to perform experiments to validate the contribution of each part of the model.

We conduct 5-fold CV of the BiTGNN model on the Enzyme database, and the experimental results are shown in Figure 2. Obviously, the accuracy of all variant models decreases to some extent, which indicates that all modules of our model are necessary and key to ensuring a good model effect. After removing the Bi-transformer module and GAT module, the performance of our model decreases greatly, which indicates that both Bi-transformer and GAT can effectively extract graph information. This also indicates that it is necessary to use the Bi-transformer module to extract cross-information of drugs and targets, and it is impossible to ignore the extraction of information node neighbor information in

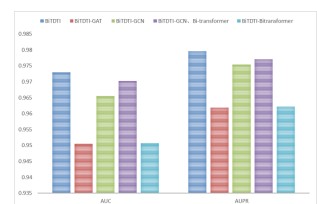

Figure 2: The result of the ablation experiments

DTI. Our model and comparison experiments without GCN show that heterogeneous graph convolution aggregates information between targets and drugs better than normal graph convolution.

Table 6: The evaluation results of BiTGNN and other baseline methods on IC's dataset.

|  | CnnDTI | Zhao et al. | PSSM+LPQ | CNNEMS | PreDTIs | DTI-HETA | Bi-TDTI |
|---|---|---|---|---|---|---|---|
| Accuracy | 0.919 | 0.8891 | 0.8601 | 0.9095 | 0.8989 | 0.9400 | **0.94344** |
| Specificity | N/A | N/A | N/A | N/A | 0.8567 | 0.944 | **0.97086** |

The best result is indicated in bold, and the second best result is marked with an underline.

### 3.5 PARAMETER ANALYSIS

In order to verify the influence of the number of common neighbors $\theta$ on model performance, we conduct experiments on a relatively large Enzyme dataset. It is a key factor that connects the similar edge $e_s$ of target and drug domains.

The results of different settings are shown in Figure 3. When $\theta$ is equal to 1, it means that if two nodes have a common neighbor, they are connected. In this case, a heterogeneous graph becomes a heterogeneous connected graph with the increase of edges, thus destroying the DTI structure and similarity. That's why the performance of $\theta$=1 is lower than others. As $\theta$ increases, the DTI plot contains more useful similar edges and the precision gets higher. However, with the increase of $\theta$, AUC and AUPR tend to rise first and then decline, as shown in Figure 2. Based on the overall conditions, $\theta = 3$ is selected for this paper.

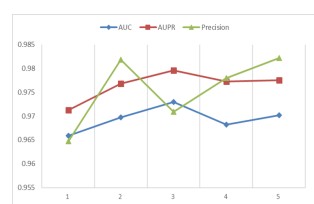

Figure 3: The effect of the common neighbor number $\theta$

### 4 CONCLUSION

Identifying potential drug-target interactions is a key task for drug discovery and drug repositioning. Although existing studies have been highly successful, improving the performance of DTI prediction remains a major challenge. In this paper, a new and comprehensive learning framework, BiTGNN, is proposed for predicting drug-target interactions, which can effectively combine the information of targets and drugs, and heterogeneous graph information. Translate drug and target features into a drug-target interaction graph. Utilize attention mechanism to assign values to edges, automatically represent the importance of edges, aggregate neighbor features using GCN, and fully refine the features of drugs and targets using Bi-transformers. We do ablation experiments to verify the importance of the module. Using cross-validation experiments on four datasets, the BiTGNN model exhibits good model performance. To verify the impact of the public neighbor threshold $\theta$ on our model, we adjust $\theta$ from 1 to 5.

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

## A  APPENDIX

You may include other additional sections here.

