# OpenReview forum: "BiTGNN: prediction of drug-target interactions based on bidirectional transformer and graph neural network on heterogeneous graph"
_ICLR.cc/2024/Conference — ICLR 2024 Conference Withdrawn Submission_

### Official Review · Reviewer_rts4 · 2023-10-28

**Soundness:** 1 poor
**Presentation:** 1 poor
**Contribution:** 1 poor
**Rating:** 1
**Confidence:** 5

**Summary:**

The authors focus on the identification of potential interactions between drugs and their targets, which is crucial for screening novel drug candidates efficiently. The authors highlight that while methods based on independent characteristics of drugs and targets have shown commendable performance, it's essential to consider drug-target interaction within a high-dimensional space. Thus, incorporating interactive information becomes rational. This study introduces BiTGNN, an improved method building upon previous work. Experimental results demonstrate that BiTGNN outperforms other state-of-the-art methods across four benchmark datasets, showing superior performance in terms of AUC, AUPR, precision, and accuracy metrics.

**Strengths:**

Please check the weaknesses below.

**Weaknesses:**

1. The novelty seems limited. It basically reuses the work of Chen et al. (2020) and fuses two models to become a bi-directional one.
2. It is not clear how the model design solves the heterogeneity issue.
3. The writing is hard to follow
4. The page limits have been exceeded.
5. No experiment comparison wrt  Chen et al. (2020)
5. Experiments table cannot be read easily.

**Questions:**

See the weaknesses above.

---

### Official Review · Reviewer_eUmF · 2023-10-29

**Soundness:** 1 poor
**Presentation:** 1 poor
**Contribution:** 1 poor
**Rating:** 1
**Confidence:** 4

**Summary:**

This manuscript aims to address the challenge of drug-target interaction (DTI) identification. The authors propose BiTGNN, which leverages graph neural network techniques, such as graph convolutional neural networks and graph attentional neural networks. Their approach incorporates pseudo-position specificity scoring matrix and drug fingerprint data to establish drug-target pairs and constructs a heterogeneous network to capture their relationship. By utilizing a bi-directional cross-attention mechanism and a feed-forward neural network, the proposed model achieves enhanced DTI prediction.

**Strengths:**

I'm really struggling to find the pros.

**Weaknesses:**

1. The article lacks innovation and feasibility in its methodology:
a) The authors seem to be unfamiliar with recent literature, as evident from their lack of knowledge on current advancements in the field.
b) The rationale behind integrating GCN and GAT is unclear.
c) The necessity of a bidirectional attention mechanism is not justified, as attention mechanisms are typically designed to address bidirectional tasks.

2. The writing style is highly disorganized. The formulas for GCN and self-attention are copied without any modification. Figure 1 is unclear, making it difficult to understand how GAT and GCN are combined. The tables are confusing, making it challenging to differentiate between different methods.

3. The data and evaluation metrics are unclear. The article does not provide information about the dataset size or the ratio of positive to negative examples. The choice of AUC and AUPR as evaluation metrics is not adequately justified. Additionally, comparing the proposed model to non-practical baseline methods such as RF and SVM raises questions about the selection of appropriate comparison benchmarks.

4. Overall, the article requires significant improvements in terms of novelty, clarity in presentation, and rigorous evaluation of the proposed model.

**Questions:**

See above.

---

### Official Review · Reviewer_UEEt · 2023-10-31

**Soundness:** 2 fair
**Presentation:** 2 fair
**Contribution:** 2 fair
**Rating:** 3
**Confidence:** 4

**Summary:**

This paper propopse a new method BiTGNN for drug-target interation prediction. The method first constructs drug-target heterogeneous graph with PsePSSM and drug fingerprints. Then it employs GCN and GAT to extract attributes of drugs and targets. Finally, the interactions are modeled by a bi-directional transformer. The paper conducts experiments on four datasets, demonstrating the superior performance of BiTGNN.

**Strengths:**

1. The topic of DTI is fundamental for drug discovery.
2. Constructing heterogeneous graph and employing GNNs for DTI is promising for this task.

**Weaknesses:**

1. The motivation and contributions are not clear. As I know, there are several studies that have constructed heterogeneous graph and have used GNNs for DTI. Thus, a comparsion to these studies should be included.

    a. Shao K, Zhang Y, Wen Y, et al. DTI-HETA: prediction of drug–target interactions based on GCN and GAT on heterogeneous graph[J]. Briefings in Bioinformatics, 2022, 23(3): bbac109.

    b. Li M, Cai X, Li L, et al. Heterogeneous Graph Attention Network for Drug-Target Interaction Prediction[C]//Proceedings of the 31st ACM International Conference on Information & Knowledge Management. 2022: 1166-1176.

    c. Yu L, Qiu W, Lin W, et al. HGDTI: predicting drug–target interaction by using information aggregation based on heterogeneous graph neural network[J]. BMC bioinformatics, 2022, 23(1): 126.

    d. Li M, Cai X, Xu S, et al. Metapath-aggregated heterogeneous graph neural network for drug–target interaction prediction[J]. Briefings in Bioinformatics, 2023, 24(1): bbac578.

2. The idea of using GNNs is not novel. As mentioned above, several studies in this field has adopted similar solutions.

3. More details for experiments should be added, including the detailed statistics of the datasets, the reasons to choose these datasets, the details about the baselines, implementation details, and hyper-parameters, etc.

4. There are many grammar mistakes. The paper should be further proofread. e.g.,

    a. 3.2 BASELINE--> 3.2 BASELINES

    b. 3.3 EXPERIMENTAL RESULT--> 3.3 EXPERIMENTAL RESULTS

    c. Figure 2: The result ... --> results

5. The presentation of the paper should also be improved. The texts in Figure 1, 2, 4 are too small.

**Questions:**

1. What are the advantages of the proposed method compared to existing studies, such as the references mentioned above?
2. How many times does the model run? Have you perform significant testing on the results?

---

### Official Review · Reviewer_Z8k5 · 2023-10-31

**Soundness:** 3 good
**Presentation:** 1 poor
**Contribution:** 2 fair
**Rating:** 3
**Confidence:** 5

**Summary:**

The paper addresses the challenge of identifying drug-target interactions (DTIs), a key area in bioinformatics crucial for drug discovery. Traditional methods for DTI identification are time-consuming and labor-intensive. The authors propose a novel model, BiTGNN (Bidirectional Transformer and Graph Neural Network), which integrates graph convolutional neural networks (GCN) and graph attentional neural networks (GAT) with a bi-directional transformer architecture. This model aims to enhance the prediction of DTIs by using a pseudo-position specificity scoring matrix (PsePSSM) and drug fingerprint data to construct a heterogeneous network. The model's performance, evaluated using Enzyme, Ion Channel (IC), G Protein-coupled Receptor (GPCR), and Nuclear Receptor (NR) datasets, reportedly surpasses existing models in several key metrics.

**Strengths:**

- The integration of GCN and GAT with a bi-directional transformer for DTI prediction is a novel approach. This multimodal method could potentially address the limitations of current models that rely on homogeneous graphs and single features.
- The use of PsePSSM and drug fingerprint data to establish drug-target pairs indicates a thorough approach to data integration, enhancing the model's ability to simulate realistic biological interactions. The authors have rigorously tested their model across multiple datasets (Enzyme, IC, GPCR, NR), providing a comprehensive validation of the model's effectiveness. The model's superior performance in AUC, AUPR, Accuracy, and Specificity metrics is commendable and suggests its potential utility in practical applications.

**Weaknesses:**

- First of all, I suggest the authors could take some time to polish this paper's formatting, including all tables and figures. The current formatting of tables and figures in the paper appears inconsistent and difficult to read. Inconsistencies in table formats and small, unclear figure legends can significantly hinder the reader's ability to understand and interpret the data and model architecture effectively.
- The model's complex architecture, particularly the use of bidirectional transformers and attention mechanisms, might pose challenges in terms of interpretability. The authors could incorporate methods to enhance the interpretability of the model, such as attention visualization or feature importance analysis. Techniques like Layer-wise Relevance Propagation (LRP) or SHAP values could be used to understand which features or parts of the graph structure most significantly influence the model's predictions, thereby providing biological insights and validation.
- The paper mentions the use of negative sampling to address the imbalance in the training data but does not delve into the specifics of how this sampling is conducted or how it impacts model performance. More details on the negative sampling strategy and its impact on model training and performance would be valuable. Exploring and comparing different approaches to handle data imbalance, such as synthetic minority over-sampling or advanced negative sampling techniques, could provide insights into the most effective strategies for DTI prediction.
- There is no mention of a robustness or sensitivity analysis in the model evaluation. Understanding how changes in input data (e.g., noise in the datasets, missing values, or erroneous entries) affect the model's predictions is crucial for evaluating its practical utility. The authors should consider including a robustness analysis, testing how the model performs with varying levels of noise or incomplete data. Additionally, a sensitivity analysis to understand how different hyperparameters (like the length of the protein sequences in PsePSSM, or the dimensionality of the drug fingerprints) impact the model's performance would be insightful.

**Questions:**

- How does the BiTGNN model perform on datasets that significantly differ from the ones used in your study, particularly those with novel or rare drug and target types?
- Could you provide more details on the computational efficiency and resource requirements of the BiTGNN model, especially in comparison to the original Transformer or GNN?